# Flexible joints performance assessment of additive manufacturing FDM 3D printed TPU

Daniel Rodríguez-Flores[1]*, Héctor Cervantes-Culebro[2,3],
J. Enrique Chong-Quero[2], Carlos A. Cruz-Villar[1]

1 Department of Electrical Engineering, CINVESTAV, Gustavo A. Madero, México City, México, 2 School of Engineering and Sciences, Tecnologico de Monterrey, Estado de México, México, 3 Bioengineering department, Imperial College of Science, Technology and Medicine, London, United Kingdom

☯ These authors contributed equally to this work.
* daniel.rodriguezf@cinvestav.mx (DR)

## Abstract

Given the increasing adoption of flexible 3D-printed joints in the field of robotics, it is essential to characterize the stiffness/spring coefficient and damping of printed specimens to understand the effects of various processing parameters and their interactions under different loading conditions. This study aims to research the effects of 3D-printed parameters and geometric dimensions, i.e, printing density, layer thickness, raster angle, length, width, and height. The Box-Benken design of experiments is conducted to obtain 44 different parametric combinations to characterize the damping and spring coefficients $C_F$, $K_F$, $C_\tau$, $K_\tau$ under a dynamic load. The damping and spring coefficients are characterized using the minimum squared method. Depending on the force decomposition, the damping and spring coefficients are different in each direction. To analyse the experimental results, a MANOVA, ANOVA analysis, and a correlation heatmap are used to show that the density, layer thickness, and width have the most influence on the damping. In contrast, raster angle has the most influence on the spring coefficient. Finally, the results show that the geometric and 3D-printing parameters play a significant role in the mechanical behaviour of flexible joints. The technology can be used in the design of robots that require energy-saving and releasing mechanisms, and to avoid the use of ball bearings under a dynamic load.

## 1 Introduction

Additive manufacturing applications of flexible materials have grown in recent years, specifically using FDM as the manufacturing technique. It is widely accepted due to its reliability, low cost, and simplicity in multiple fields. Some of the main fields benefiting from this technology include biomedical [1,2], automotive [3], aerospace [4], communications [5,6], the food sector [7], and education [8].

**Data availability statement:** The complete dataset and the design of experiment can be found in the following url: https://simtk.org/projects/flex-tpu.

**Funding:** A part of this work funding was provided by CONAHCYT. The author, Héctor Cervantes-Culebro, would like to acknowledge the financial support of NOVUS (Grant number N22-302) and Writing Lab, Institute for the Future of Education, Tecnologico de Monterrey, Mexico, in the production of this work. The funders had no role in the study design, data collection and analysis, decision to publish.

**Competing interests:** The authors have declared that no competing interests exist.

In the FDM process, the filament material is fed into a heated nozzle, extruded in a semi-liquid state, and deposited onto the previously printed layer on the build platform, where it quickly solidifies and bonds to the previous layer. After each layer is deposited, the extruder moves upwards along the z-axis by the height of one layer thickness, repeating the process until the component is completed [9].

However, the layer-by-layer nature of the FDM manufacturing process results in limited surface and mechanical characteristics that can lead to undesired porosity, delamination, stress concentrators, reduced mechanical performance, or premature failure. The combination of printing parameters modifies the performance of the flexible joints [10], [11].

Given the changes in properties caused by the 3D-printing process, it is essential to characterize the mechanical performance of printed specimens under various loadings. Several studies have analyzed the mechanical response of flexible 3D-printed polymers, providing insights into their performance. For instance, Sadaghian et al. [12] investigated the behavior of 15 different 3D-printed materials under monotonic torsion, observing that failure occurred in most specimens except for TPU. Other types of studied loads include compression [13], tension [14], and deflection.

The selected application for this study is 3D-printed flexible joints for robots. Some examples of flexible joints in robotics are for Quadruped Robots to enhance its impact buffering capability as well as reducing energy consumption [15], robot arms that mimic the free movement of the human arm with guarantee transient performance [16], miniaturized soft robots for in vitro or in vivo applications [17], Soft artificial muscles for lifting objects mimicking chameleon tongues [18] and exoskeletons for motion assistance in rehabilitation [19].

Various studies have employed different fabrication techniques to characterize further the impact resistance of 3D-printed flexible parts made from TPU under different loading conditions, such as tensile, torsion, fatigue, and impact. These include selective laser sintering to examine TPU powder which had been cryogenically milled in two different sizes [20], evaluation an optimization of the selective laser sintering process [21], mechanical characterization of 3D-printed polymers [22], FDM [23], fused filament fabrication [24], and investigations into the influence of different 3D-printing parameters into the applicability of said materials [25]. Understanding these factors is crucial for optimizing the mechanical performance of flexible joints in industrial robotic applications.

Due to the nature of the FDM process, the resulting parts tend to have anisotropic mechanical properties [26]. Many testing runs and thorough analyses are required to explore all parameter combinations and understand the effects of various processing parameters and their interactions under different loading conditions. Researchers often utilize Design of Experiments (DOE) techniques to optimize the process while minimizing the number of experimental runs and analysis procedures.

Some of the most commonly used DOE methodologies include mixed factorial [27], Taguchi [28], Box-Behnken [29], and different optimality criteria such as A-optimality, C-optimality, D-optimality, E-optimality, S-optimality, T-optimality, G-optimality, I-optimality, and V-optimality [30–32]. Additionally, techniques like the Central Composite Design method [33], Central Composite Inscribed Design, and

Central Composite Face-Centered Design [34] are frequently employed. By incorporating these methodologies, researchers can identify the optimal combination of FDM process variables to enhance the mechanical properties of flexible joints in robotic applications. The chosen methodology in this work for the experiment is the Box-Behnken design, given the number of parameters and levels for each parameter.

DOE techniques are often coupled with statistical optimization methods to refine process optimization further. These include Genetic Algorithms, Artificial Neural Networks (ANN) [35], Fuzzy Logic [36], Gray Relational Analysis [37], Response Surface Methods [38], Bacterial Foraging Optimization [39], and Particle Swarm Optimization [40].

Compared to past research where the characterization of the material is under a monotonic or continuous load. The main objective is characterize the stiffness/spring coefficient and damping of printed specimens under a dynamic load, followed by researching the effects of 3D-printed parameters and geometric dimensions in the mechanical characteristics of flexible joints. A combination of torsional and flexural loads are applied to FDM TPU parts, no other studies using flexural loads were found. To obtain the stiffness/spring and damping coefficients six process parameters in the manufacturing process are considered.

The primary application of these parts is for robots' flexible joints. A standard approximation for modeling flexible joints is using the pseudo-rigid body model [41], which represents the flexible joint as a standard joint with a torsional spring. However, this model does not account for the damping phenomena present in 3D-printed polymer flexible joints. To address this limitation, the resulting angular deflection, its derivatives, forces, and moments of the experiment are obtained; these variables are required to obtain the mass-spring-damper coefficients that approximate the flexible joint, enabling a more accurate representation of the joint's behavior.

This article is divided into five sections. The second section presents the proposed mass-spring-damper model. In the third section, the research methodology is presented, including the chosen material and manufacturing method for the test specimens, the DOE methodology chosen to simplify the experiment, characterize coefficients per axis, and the model validation methodology. The fourth section presents the proposed experiment's results, analysis, and discussion. In the fifth section, the conclusion and references are presented.

## 2 Proposed mass-spring-damper model

The most commonly used model for flexible joints in mechanisms is the pseudo-rigid-body model [42]. The pseudo-rigid-body model approximates the behaviour of the flexible joints as a spring. The approximation is limited to a region of operation. Assuming the internal lattice structures are homogeneous, the behaviour of the flexible joint can be approximated by the proposed mass-spring-damper model.

For the flexible joint subjected to linear and angular deformations, Eq (1), and (3) are proposed.

$$\mathbf{M}\ddot{r}(t) + \mathbf{C_F}\dot{r}(t) + \mathbf{K_F}r(t) = F(t) \tag{1}$$

$$\mathbf{M} = \begin{bmatrix} m_x & 0 & 0 \\ 0 & m_y & 0 \\ 0 & 0 & m_z \end{bmatrix}, \quad \mathbf{C_F} = \begin{bmatrix} c_x & 0 & 0 \\ 0 & c_y & 0 \\ 0 & 0 & c_z \end{bmatrix}, \quad \mathbf{K_F} = \begin{bmatrix} k_x & 0 & 0 \\ 0 & k_y & 0 \\ 0 & 0 & k_z \end{bmatrix},$$

$$r(t) = \begin{bmatrix} x(t) \\ y(t) \\ z(t) \end{bmatrix}, \quad F(t) = \begin{bmatrix} F_x(t) \\ F_y(t) \\ F_z(t) \end{bmatrix} \tag{2}$$

$$\mathbf{I}\dot{\omega}(t) + \mathbf{C_\tau}\omega(t) + \mathbf{K_\tau}\Theta(t) = \tau(t) \tag{3}$$

$$\mathbf{I} = \begin{bmatrix} I_\theta & 0 & 0 \\ 0 & I_\phi & 0 \\ 0 & 0 & I_\Psi \end{bmatrix}, \quad \mathbf{C}_\tau = \begin{bmatrix} c_\theta & 0 & 0 \\ 0 & c_\phi & 0 \\ 0 & 0 & c_\Psi \end{bmatrix}, \quad \mathbf{K}_\tau = \begin{bmatrix} k_\theta & 0 & 0 \\ 0 & k_\phi & 0 \\ 0 & 0 & k_\Psi \end{bmatrix},$$

$$\Theta(t) = \begin{bmatrix} \theta(t) \\ \phi(t) \\ \Psi(t) \end{bmatrix}, \quad \tau(t) = \begin{bmatrix} \tau_\theta(t) \\ \tau_\phi(t) \\ \tau_\Psi(t) \end{bmatrix} \tag{4}$$

where $\mathbf{M}$ is the matrix of mass of the specimen, $\mathbf{I}$ is the inertia matrix, $\mathbf{C_F}$, $C_\tau$ are the damping coefficients matrices, $\mathbf{K_F}$, $\mathbf{K}_\tau$ are the springs coefficients matrices, $F$, $\tau$ are the applied forces or moments vectors, $\Theta$, $\omega$, $\dot{\omega}$ are the angular positions and its' first and second time derivatives vectors respectively.

The values in matrices $\mathbf{M}$, $\mathbf{C_F}$, $\mathbf{K_F}$, $\mathbf{I}$, $\mathbf{C}_\tau$, $\mathbf{K}_\tau$ are identified with experimental data using the closed form solution of the least squared problem.

$$\hat{\beta} = \left(\mathbf{X^T X}\right)^{-1} \mathbf{X^T y} \tag{5}$$

where $\mathbf{X} \in \mathbb{R}^{3 \times n}$ is the input variables, $n$ is the number of data points in each experiment, $\mathbf{y} \in \mathbb{R}^{3 \times n}$ are the output variables and $\hat{\beta}$ are the estimated coefficients. In the case of the proposed experiment the inputs are the positions $(r, \theta)$, speed $(\dot{r}, \omega)$, and velocities $(\ddot{r}, \dot{\omega})$, and the outputs are the forces and torques $(F, \tau)$.

# 3 Research methodology

## 3.1 Material and methods

In this study, rectangular cross-section test specimens were manufactured using different printing and geometric parameters. The 3D printed specimens for this paper are made using a commercial 95A TPU made by Creality, and the 3D printer used is an Ender 3 V2 FDM printer.

The process parameters and their levels considered for the manufacturing process are presented in Tables 1 and 2. Fig 1 presents the printer, printed specimens with the different levels of their parameters, and the material used. The geometry chosen for the test specimens corresponds to a rectangular cross-section; the whole geometry is shown in Fig 2. This geometry simplifies the number of variables required to be analyzed, reducing the possible combinations

**Table 1**. Process parameters.

| Parameters | Value |
|---|---|
| Nozzle diameter (mm) | 0.4 |
| Bed temperature (°C) | 60 |
| Liquefier temperature (°C) | 210 |
| Infill pattern | Cubic |
| Shell Thickness (mm) | 0.8 |

**Table 2**. Parameters and their levels.

| Parameters | Levels | | |
|---|---|---|---|
| | 1 | 2 | 3 |
| Printing density (%) | 20 | 60 | 100 |
| Length $l$ (mm) | 18 | 24 | 30 |
| Width $b$ (mm) | 2 | 4.5 | 7 |
| Height $h$ (mm) | 4 | 9 | 14 |
| Layer thickness (mm) | 0.12 | 0.16 | 0.2 |
| Raster angle (°) | 0 | 45 | – |

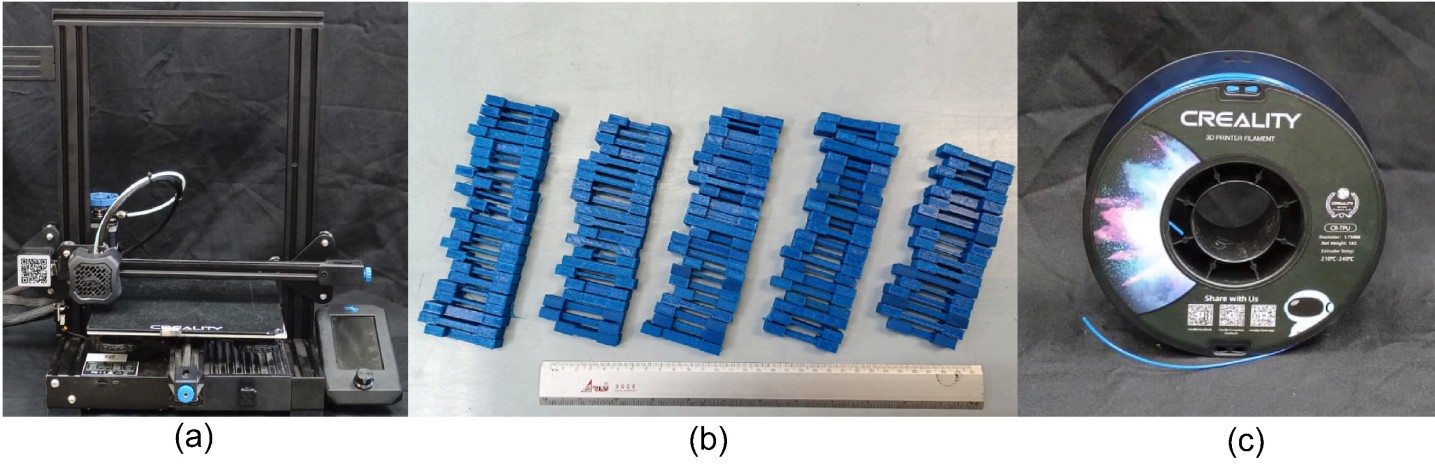

(a)                                      (b)                                      (c)

**Fig 1. Printer and printed parts.** (a) Creality Ender 3 V2 3D printer. (b) 3D printed TPU test specimens.

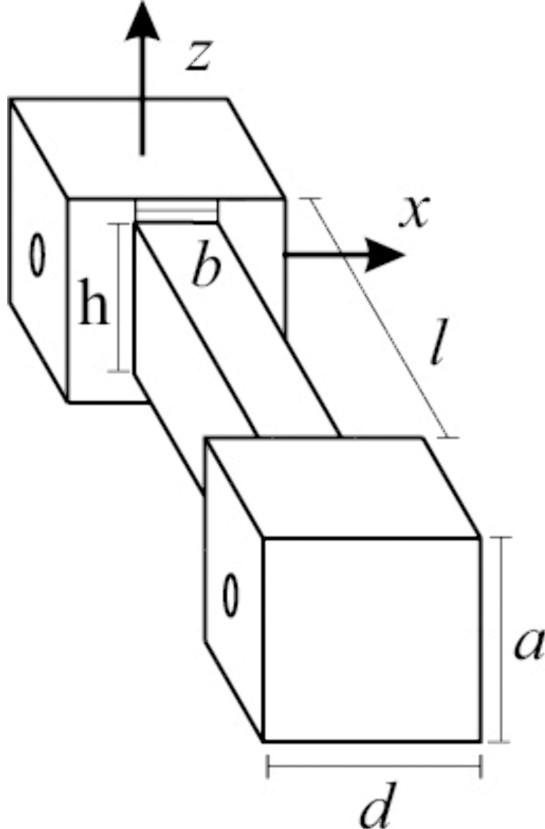

**Fig 2. Test specimen geometry.** The cross-section of the test specimen corresponds to a rectangle of base *b* and height *h*; the length of the specimen is *l*. The extremes of the test specimen are used for clamping to the testing machine and maintain the exact dimensions *a* and *b* across all test specimens.

required for the tests. The extremes of the test specimen are for clamping, dimensions *a* and *d* of both geometries are the same across all test specimens. The 3D-printing variable parameters were chosen assuming their maximum impact given similar research on tension and compression conditions [43–45].

Taking into consideration the 3D printing process parameters and the geometric parameters of interest and their levels in Table 2, the experiment is designed to obtain the required information to approximate the test specimen model as a spring-damper coefficient.

### 3.2 Design of experiment

A periodical combined flexural and torsional load is applied to the test specimen, and the resulting deformations are used to assess the mechanical response and properties at different frequencies. Simulating a load experienced by a robot's flexible joint. The testing machine is shown in Fig 3. It is comprised of a hub marked in green, clamped at one end of the test specimen marked in blue that moves in a sinusoidal motion at a 45° angle about the test specimen, generating the desired combination of flexural and torsional loads. The other end of the test specimen is attached to a UFACTORY six-axis force torque sensor at the end effector of a UFACTORY xArm 6 robot arm, marked in red.

The loads are shown by the diagram in Fig 4, where the angular deformations are determined by $\theta$, $\phi$, and $\Psi$ depending on the direction of deformation.

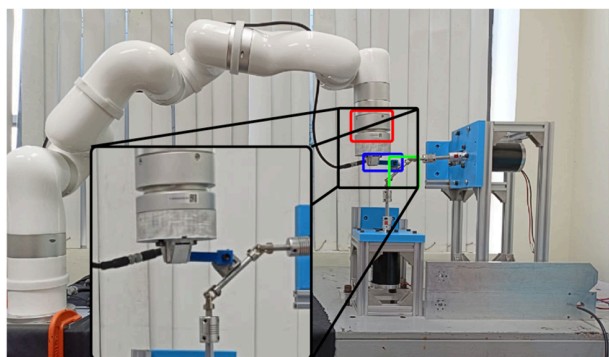

**Fig 3. Testing machine.** The test specimen marked in blue is fixed to the hub marked in green that applies the combination flexural and torsional load, the other end is fixed to the six-axis force torque sensor attached at the free end of the robot arm marked in red. The force torque sensor is marked on red, the test specimen is marked on blue and the hub is marked on green.

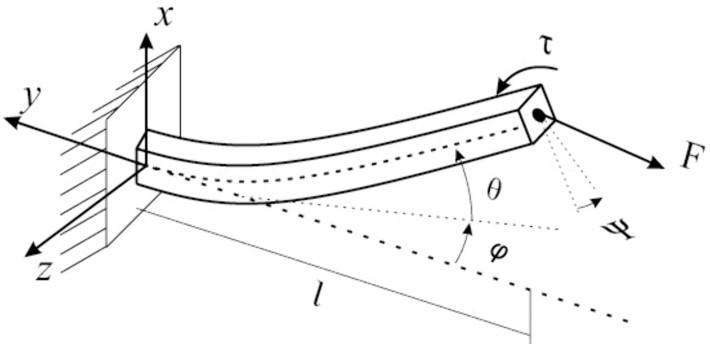

**Fig 4. Test specimen applied loads.** The torsional and flexural loads applied to the test specimen are described by the variables $\tau$, $F$, and the resulting angular deformations defined by $\theta$, $\phi$, and $\Psi$, depending on the direction of deformation.

The angular position of the hub, motor currents, robot positions, end effector positions, forces, and moments are logged. The hub moves in a sinusoidal pattern with an amplitude of 1 *rad*, generating a load that is applied periodically at an increasing frequency rate starting from 0.2 *Hz* up to 1.85 *Hz* at the end of the experiment. The frequency rate is picked according to similar research in trajectory planning for robots [46]. A complete run of the experiment lasts for 2200 seconds.

The experiment is designed using the Minitab 22 tools of response surface design Box-Behnken method, where Table 2 presents the variables of the current research and their intervals. Levels 1 and 3 are chosen by design, and level 2 is generated by the DOE Minitab tool. The least squares technique is used to fit the dynamic mass-spring-damper model shown in Eqs (1) and (2) for the linear deformations and Eqs (3) and (4) for angular deformations. This model assumes that mass $m$, spring coefficient $c$, and damping coefficient $k$ differ depending on the axis. These conditions are given by the geometry of the test specimen, whose cross section is not symmetric, the anisotropic properties of the material and manufacturing method, the natural frequencies, and the settling time that differ depending on the direction [47].

The TPU test specimens were manufactured based on the DOE using the Box-Behnken design. The FDM printing parameters and specimen dimensions according to Table 2 are: printing density, layer thickness, raster angle, length, width, and height. The layer thickness, and raster angle are considered categorical factors for DOE. One of the advantages of using DOE is reducing the number of tests and costs, and the possibility of optimizing the parameters and estimating the effectiveness of each factor [48]. Therefore, 160 specimens with different parameters were printed to study the response against a combined torsional and flexural load. In order to increase the accuracy of the results and eliminate the effect of printing quality, simple geometries were used for the specimens as shown in Fig 2, where only the three principal dimensions are highlighted, generating a test specimen with a rectangular cross section.

Once the experiments are conducted, the resulting coefficients are used as the dependent variables of a Multivariate analysis of variance (MANOVA) and an analysis of variance (ANOVA) to determine the correlation between each of the variable process parameters and combinations between them. The coefficients are also validated using the following methodology.

## 4 Results and discussion

All the experiments were performed using the experimental Box-Behnken design. To ensure the validity of the datasets the results of the 160 test specimens were filtered and cleaned. Sensor errors result in zeroes or NaN values making them useless for the analysis. In the case of repeated data, the wall thickness is 1 mm so when the variable width $b$ is 2 mm the printing density becomes 100 % resulting in an overlap of the data, the outliers were detected using DBScan. The parameters used for the DBScan are $\epsilon = 0.5$ and a minimum cluster of 5 points. The data cleaning results in 44 usable datasets. The experimental results of such datasets are shown in Table 3, the experiments were performed only once. The calculated damping and spring coefficients are different depending on the direction of the deformations.

The resulting spring and damping coefficients show that this type of flexible joint can be used in applications that require energy-saving and release mechanisms. The damping coefficients allow for substitution of ball bearing joints for flexible joints at the expense of reduced angular motion.

To ensure statistical significance, the relation of each manufacturing parameter and the validity of the resulting damping and spring coefficients are used as factors in the analysis.

### 4.1 MANOVA and ANOVA

For the MANOVA and ANOVA analysis, the output parameters are defined as a function of the printing density, layer thickness, raster angle, length, width, and height. The equations fitted by the least squares technique are the first-order differential equations Eq (1) and Eq (3), where $\mathbf{M}$, $\mathbf{I}$, $\mathbf{C_F}$, $\mathbf{C_\tau}$, and $\mathbf{K_F}$, $\mathbf{K_\tau}$ are the constant coefficients of linear, quadratic terms, and cubic terms, respectively.

**Table 3**. Experimental data.

| Run No. | Printing density (%) | Length (mm) | Height (mm) | Width (mm) | Layer thickness (mm) | Raster angle (°) | Mean X force (N) | $c_x$ (Ns/mm) | $k_x$ (N/mm) |
|---|---|---|---|---|---|---|---|---|---|
| 1 | 20 | 18 | 9 | 4.5 | 0.12 | 45 | 1.53 | $2.24 \times 10^{-3}$ | 0.679 |
| 2 | 20 | 24 | 9 | 2 | 0.12 | 45 | 1.17 | $1.91 \times 10^{-3}$ | 0.920 |
| 3 | 20 | 24 | 9 | 7 | 0.12 | 45 | 1.34 | $3.40 \times 10^{-3}$ | 0.651 |
| 4 | 20 | 24 | 4 | 4.5 | 0.12 | 45 | 1.98 | $1.15 \times 10^{-3}$ | 1.196 |
| 5 | 20 | 24 | 14 | 4.5 | 0.12 | 45 | 1.40 | $2.47 \times 10^{-3}$ | 0.616 |
| 6 | 60 | 18 | 9 | 2 | 0.12 | 45 | 1.30 | $2.56 \times 10^{-3}$ | 0.996 |
| 7 | 60 | 24 | 4 | 2 | 0.16 | 45 | 0.74 | $1.15 \times 10^{-3}$ | 0.793 |
| 8 | 60 | 24 | 4 | 7 | 0.16 | 45 | 1.33 | $3.22 \times 10^{-3}$ | 0.838 |
| 9 | 60 | 18 | 4 | 4.5 | 0.16 | 45 | 1.44 | $3.63 \times 10^{-3}$ | 0.934 |
| 10 | 60 | 18 | 14 | 4.5 | 0.16 | 45 | 1.17 | $1.00 \times 10^{-3}$ | 0.541 |
| 11 | 60 | 18 | 9 | 2 | 0.16 | 45 | 1.42 | $3.16 \times 10^{-3}$ | 0.995 |
| 12 | 60 | 30 | 9 | 2 | 0.16 | 45 | 0.93 | $1.59 \times 10^{-3}$ | 0.615 |
| 13 | 60 | 18 | 9 | 7 | 0.16 | 45 | 0.95 | $1.86 \times 10^{-3}$ | 0.483 |
| 14 | 60 | 30 | 9 | 7 | 0.16 | 45 | 1.02 | $2.01 \times 10^{-3}$ | 0.550 |
| 15 | 100 | 30 | 9 | 4.5 | 0.2 | 45 | 1.22 | $4.91 \times 10^{-4}$ | 0.515 |
| 16 | 60 | 24 | 14 | 7 | 0.2 | 45 | 0.72 | $9.08 \times 10^{-4}$ | 0.246 |
| 17 | 60 | 18 | 4 | 4.5 | 0.2 | 45 | 1.60 | $1.32 \times 10^{-3}$ | 0.728 |
| 18 | 60 | 18 | 9 | 7 | 0.2 | 45 | 1.05 | $1.43 \times 10^{-3}$ | 0.596 |
| 19 | 20 | 18 | 9 | 4.5 | 0.12 | 0 | 1.48 | $2.28 \times 10^{-3}$ | 0.780 |
| 20 | 20 | 30 | 9 | 4.5 | 0.12 | 0 | 1.18 | $2.67 \times 10^{-3}$ | 0.724 |
| 21 | 60 | 24 | 4 | 2 | 0.12 | 0 | 1.41 | $5.14 \times 10^{-4}$ | 2.092 |
| 22 | 20 | 24 | 9 | 2 | 0.12 | 0 | 0.87 | $1.25 \times 10^{-3}$ | 0.777 |
| 23 | 100 | 24 | 9 | 2 | 0.12 | 0 | 1.12 | $1.92 \times 10^{-3}$ | 0.837 |
| 24 | 20 | 24 | 9 | 7 | 0.12 | 0 | 1.41 | $2.76 \times 10^{-3}$ | 0.705 |
| 25 | 20 | 24 | 4 | 4.5 | 0.12 | 0 | 1.47 | $2.32 \times 10^{-3}$ | 0.649 |
| 26 | 100 | 24 | 4 | 4.5 | 0.12 | 0 | 1.28 | $3.09 \times 10^{-3}$ | 0.789 |
| 27 | 20 | 24 | 14 | 4.5 | 0.12 | 0 | 1.31 | $2.97 \times 10^{-3}$ | 0.587 |
| 28 | 20 | 30 | 9 | 4.5 | 0.16 | 0 | 1.37 | $3.60 \times 10^{-3}$ | 0.795 |
| 29 | 60 | 24 | 14 | 2 | 0.16 | 0 | 1.27 | $3.45 \times 10^{-3}$ | 0.859 |
| 30 | 20 | 24 | 9 | 2 | 0.16 | 0 | 1.09 | $1.77 \times 10^{-3}$ | 0.738 |
| 31 | 60 | 30 | 4 | 4.5 | 0.16 | 0 | 1.30 | $4.52 \times 10^{-3}$ | 0.903 |
| 32 | 60 | 18 | 14 | 4.5 | 0.16 | 0 | 1.15 | $1.07 \times 10^{-3}$ | 0.407 |
| 33 | 60 | 30 | 14 | 4.5 | 0.16 | 0 | 1.29 | $2.25 \times 10^{-3}$ | 0.650 |
| 34 | 20 | 24 | 4 | 4.5 | 0.16 | 0 | 1.42 | $3.98 \times 10^{-3}$ | 0.910 |
| 35 | 20 | 30 | 9 | 4.5 | 0.16 | 0 | 1.85 | $2.78 \times 10^{-3}$ | 0.815 |
| 36 | 60 | 24 | 9 | 4.5 | 0.16 | 0 | 1.38 | $1.75 \times 10^{-3}$ | 0.614 |
| 37 | 60 | 24 | 4 | 7 | 0.2 | 0 | 1.32 | $2.98 \times 10^{-3}$ | 0.657 |
| 38 | 20 | 24 | 9 | 7 | 0.2 | 0 | 1.44 | $2.21 \times 10^{-3}$ | 0.454 |
| 39 | 60 | 18 | 4 | 4.5 | 0.2 | 0 | 1.39 | $2.18 \times 10^{-3}$ | 0.820 |
| 40 | 60 | 30 | 4 | 4.5 | 0.2 | 0 | 1.25 | $2.70 \times 10^{-3}$ | 0.691 |
| 41 | 60 | 18 | 14 | 4.5 | 0.2 | 0 | 1.10 | $2.97 \times 10^{-3}$ | 0.642 |
| 42 | 100 | 24 | 4 | 4.5 | 0.2 | 0 | 1.25 | $2.00 \times 10^{-3}$ | 0.752 |
| 43 | 60 | 30 | 9 | 7 | 0.2 | 0 | 0.93 | $1.51 \times 10^{-3}$ | 0.406 |
| 44 | 60 | 24 | 9 | 4.5 | 0.2 | 0 | 1.27 | $2.50 \times 10^{-3}$ | 0.606 |

The MANOVA is used to test the impact of the independent variables on the dependent spring and damping coefficients simultaneously. For a deeper analysis, the ANOVA is used to estimate the importance of each 3D-printing and geometric parameter on each of the output responses and assess the fitness of each of the presented coefficients. The results of the MANOVA for the *x* associated coefficients are displayed in Table 4. The **Statistic** column corresponds to the statistical tools used, Pillai, Wilks, Hotelling, and Roy (P, W, H, R). Columns five and six **DF1** and **DF2**, correspond to the

**Table 4. Multivariate analysis of variance for *x* coefficients.**

| Source | Statistic | F | RSS | DF1 | DF2 | P-value | Contribution (%) |
|---|---|---|---|---|---|---|---|
| (Intercept) | P,W,H,R | 49.494 | 0.572 | 1 | 37 | $2.544 \times 10^{-8}$ | 11.781 |
| RasterAngle | P,W,H,R | 0.331 | 0.009 | 1 | 37 | 0.568 | 0.183 |
| Layer | P,W,H,R | 4.458 | 0.108 | 1 | 37 | **0.042** | 2.214 |
| Width | P,W,H,R | 9.790 | 0.209 | 1 | 37 | **0.003** | 4.308 |
| Height | P,W,H,R | 16.591 | 0.310 | 1 | 37 | **0.000** | 6.374 |
| Length | P,W,H,R | 0.238 | 0.006 | 1 | 37 | 0.628 | 0.132 |
| Density | P,W,H,R | 0.016 | 0.000 | 1 | 37 | 0.901 | 0.009 |

number of degrees of freedom in the numerator and denominator, respectively. The eighth column is the **Contribution**, measures of how much each independent variable contributes to the multivariate separation between groups.

The MANOVA analysis shows the significance of the presented model, where three of the six variables, layer thickness, width, and length, have a P-value $\leq 0.05$, and the higher contributions among the parameters. The highest contribution is given by the intersection, as expected. To help assess whether mean differences among groups on a combination of dependent variables are statistically significant.

The effect of each independent variable on the damping and spring coefficients is shown in Figs 5-8 for each of the linear deformations and Figs 9-12 for the angular deformations. In the case of Figs 5-8. There is no change in the damping coefficient with the change of the raster angle, but on the spring, a higher coefficient is observed with 0° raster angle. For the *x* axis, a finer layer thickness increases both the damping and spring effect; the width, height, and length have an inverse effect on the spring, where the thinner the test specimen, the less effect, and for the damping, the thicker, the more damping. The printing density of the specimen has the most influence on the damping but almost no influence on the spring. The influence of the geometric variables is associated with the capacity of the test specimen to bend. When

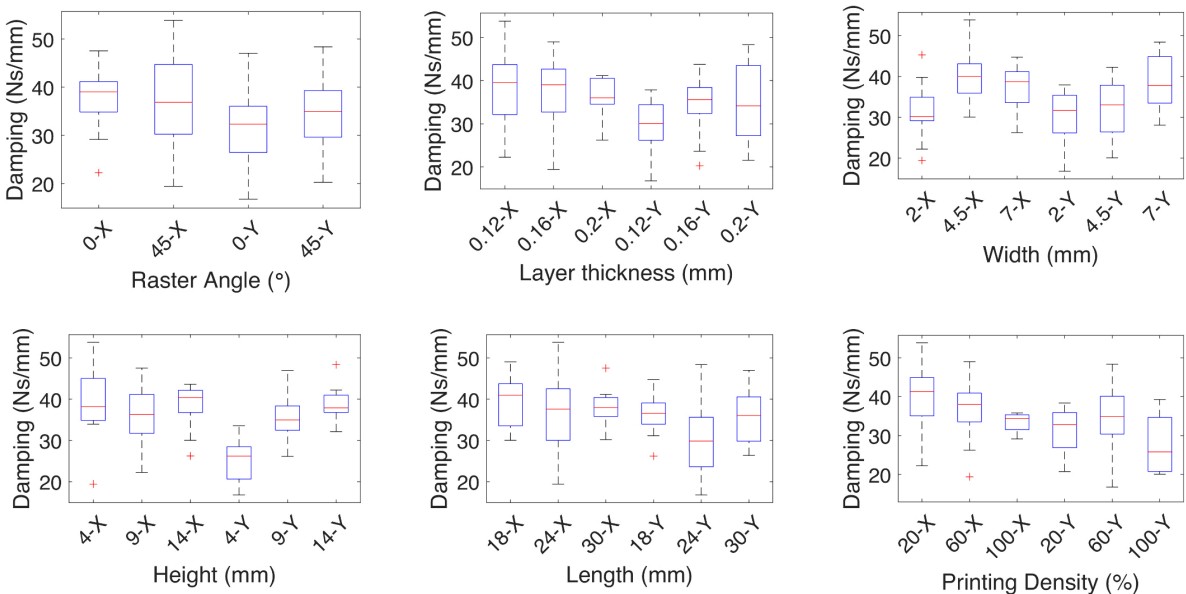

**Fig 5. Effect of the independent variables on the damping coefficient for linear deformations.** The effect of each independent variable on the damping coefficients are shown for the linear deformations *X*, and *Y*, the medians are marked by red lines and show the increase in stiffness at a 45° raster angle, layer thickness increases the damping, influence of the geometric variables is associated with the capacity of the test specimen to bend, the printing density of the specimen has the most influence on the damping.

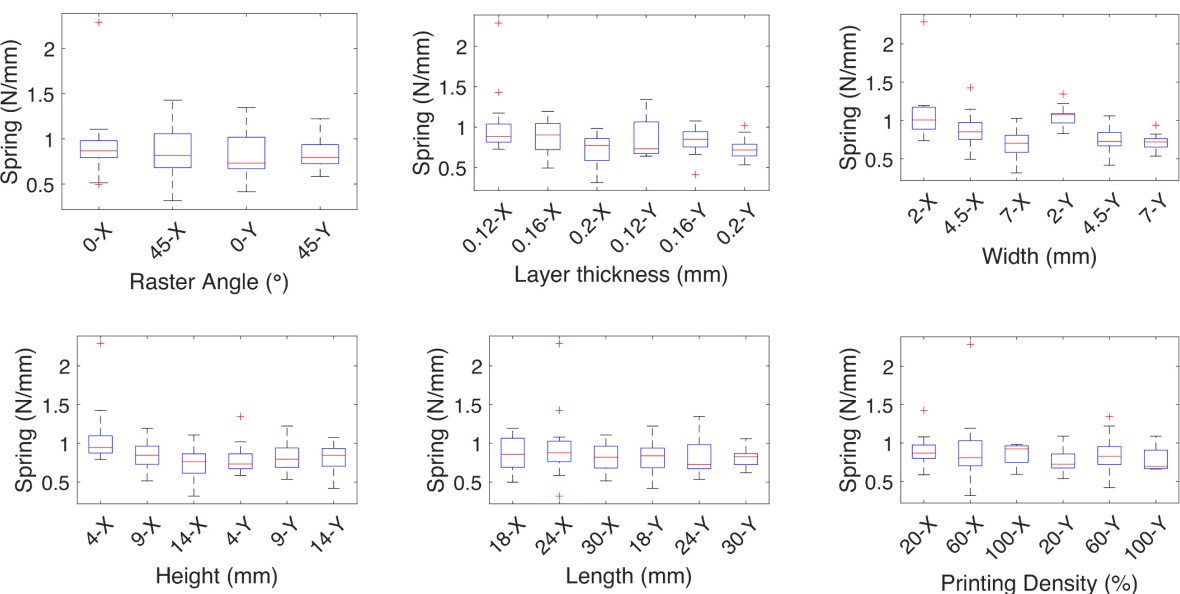

**Fig 6**. **Effect of the independent variables on the spring coefficient for linear deformations.** The effect of each independent variable on the spring coefficients are shown for the linear deformations *X*, and *Y*, the medians are marked by red lines, the geometric variables width, height, and length have an inverse effect on the spring where the thinner the test specimen, the less effect, the spring coefficient increases with the layer thickness.

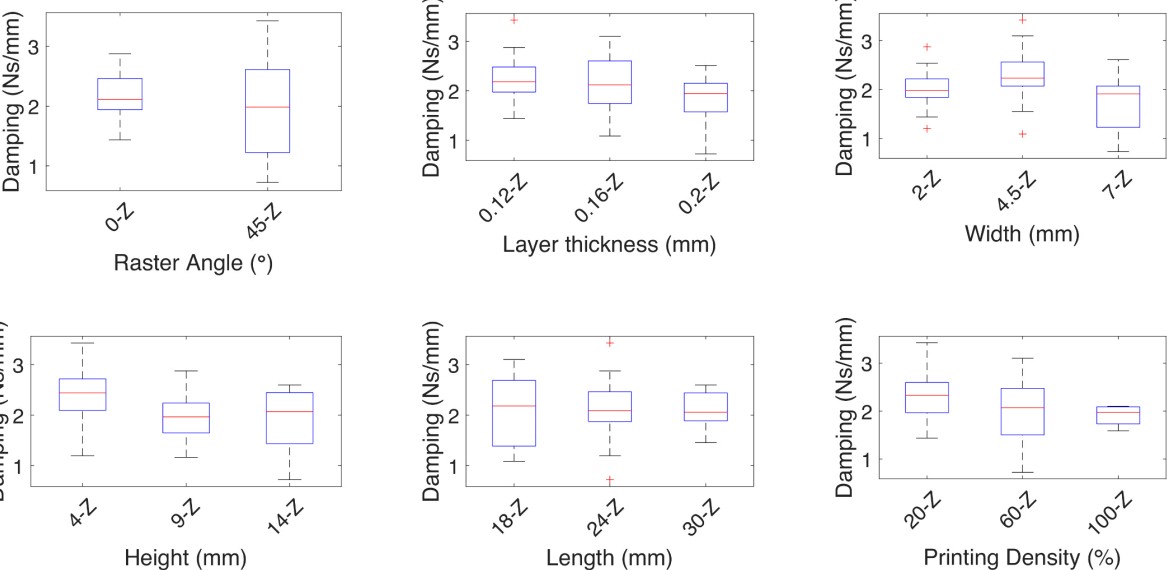

**Fig 7**. **Effect of the independent variables on the damping coefficient for linear deformation *Z*.** The effect of each independent variable on the damping coefficients are shown for the linear deformation *Z*, the medians are marked by red lines and show the increase in damping at a 0° raster angle, and a low sensibility to the other variables given the medians are similar.

the width is 2 mm, the density parameter becomes irrelevant. This is due to the wall thickness being 0.8 mm, leaving room for 0.2 mm of space, which the resolution of the 3D printer can not guarantee on the wall thickness parameter.

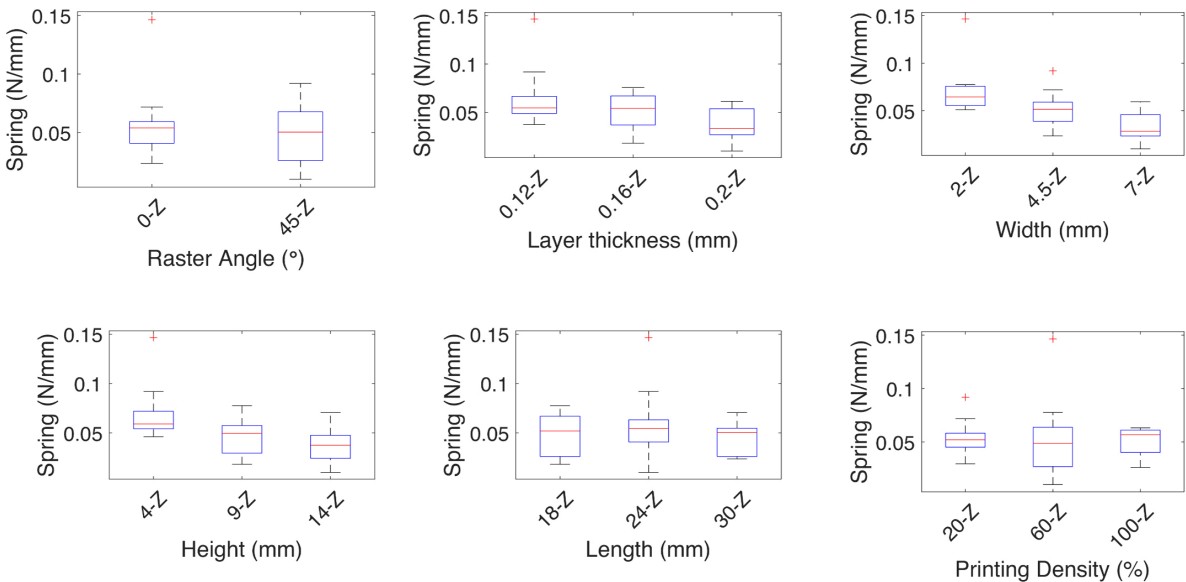

**Fig 8**. **Effect of the independent variables on the spring coefficient for linear deformation Z.** The effect of each independent variable on the spring coefficients are shown for the linear deformation Z, the medians are marked by red lines, the geometric variables width, and height have the most effect on the spring coefficient.

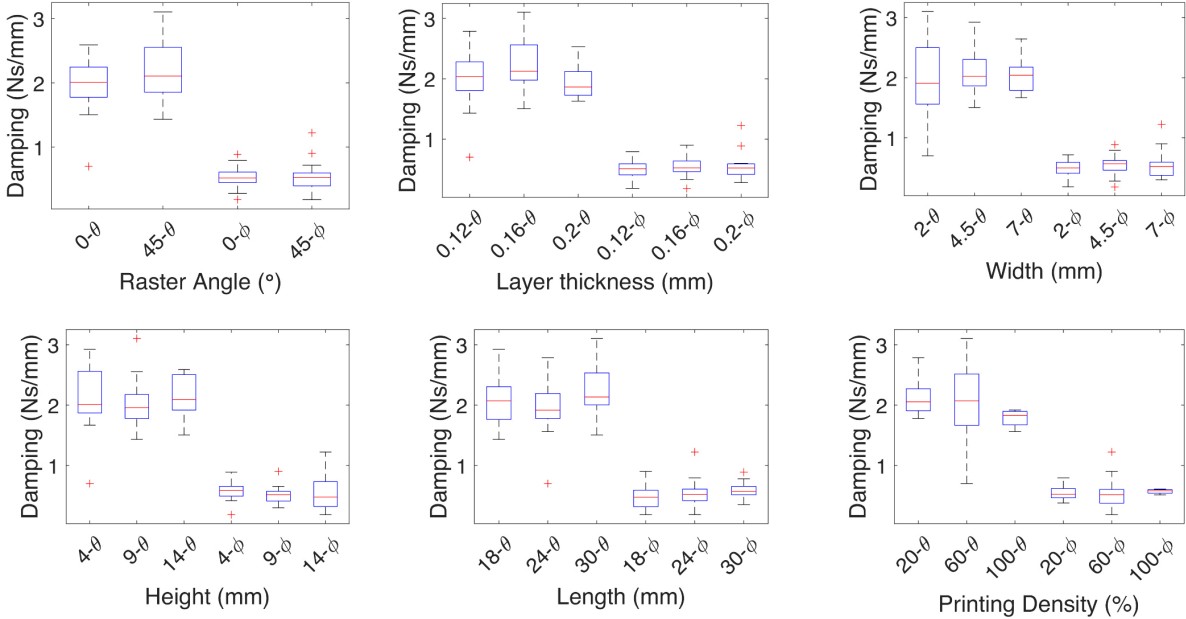

**Fig 9**. **Effect of the independent variables on the damping coefficient for angular deformations.** The effect of each independent variable on the damping coefficients are shown for the angular deformations $\theta$, and $\phi$, the medians are marked by red lines, has higher medians as the width and height values increase.

In the case of the coefficients for the *y* axis deformation, for the damping, the 45° raster angle has a higher median than the 0° raster angle, which has more variability. The higher the layer thickness, the higher the damping coefficient,

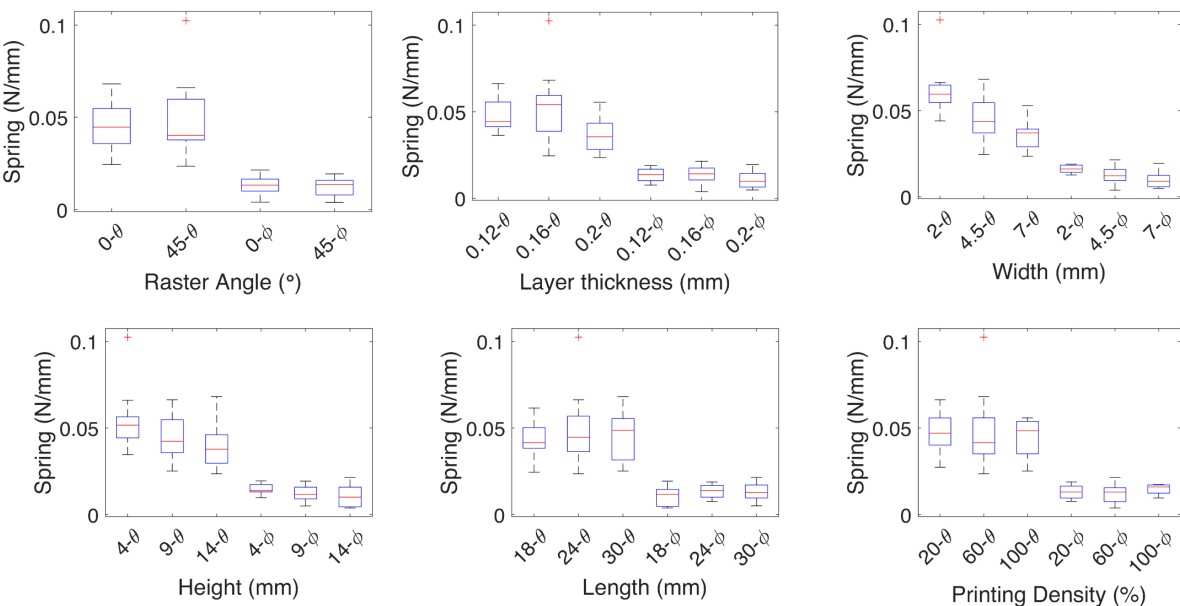

**Fig 10**. **Effect of the independent variables on the spring coefficient for angular deformations.** The effect of each independent variable on the spring coefficients are shown for the angular deformations $\theta$, and $\phi$,, the medians are marked by red lines, the spring coefficient at 0° raster angle is higher and has less variability, has lower medians as the width, length, and height increase.

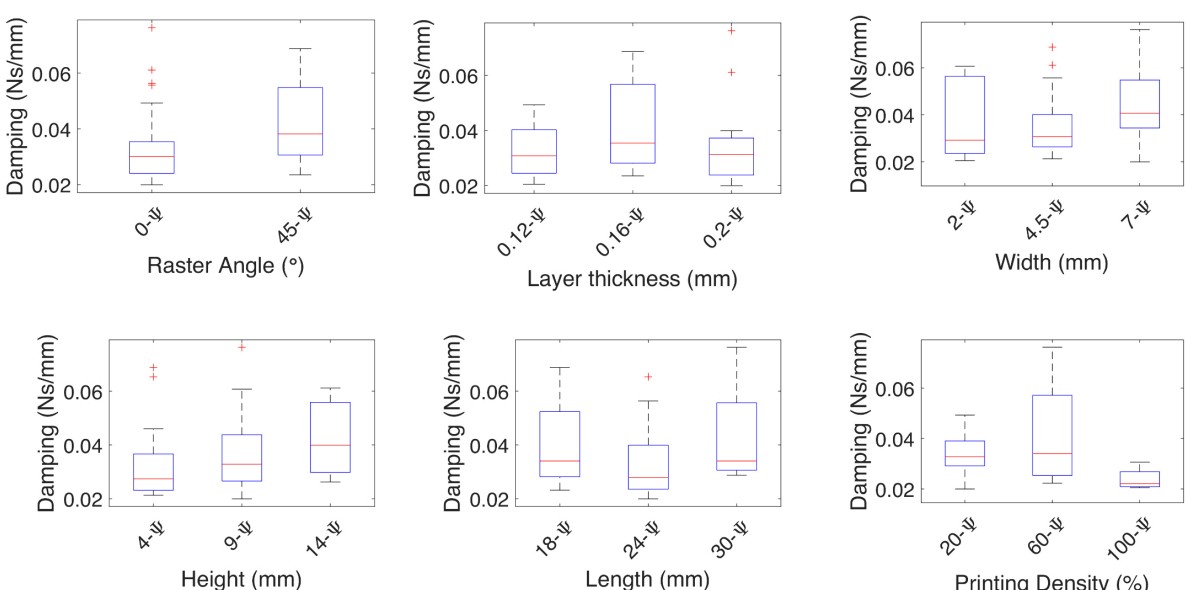

**Fig 11**. **Effect of the independent variables on the damping coefficient for angular deformations.** The effect of each independent variable on the damping coefficients are shown for the angular deformation $\Psi$, the medians are marked by red lines, the spring coefficient at 45° raster angle is higher.

meaning higher energy dissipation; a similar effect is observed with the length. The spring coefficient increases with the layer thickness and decreases with the width.

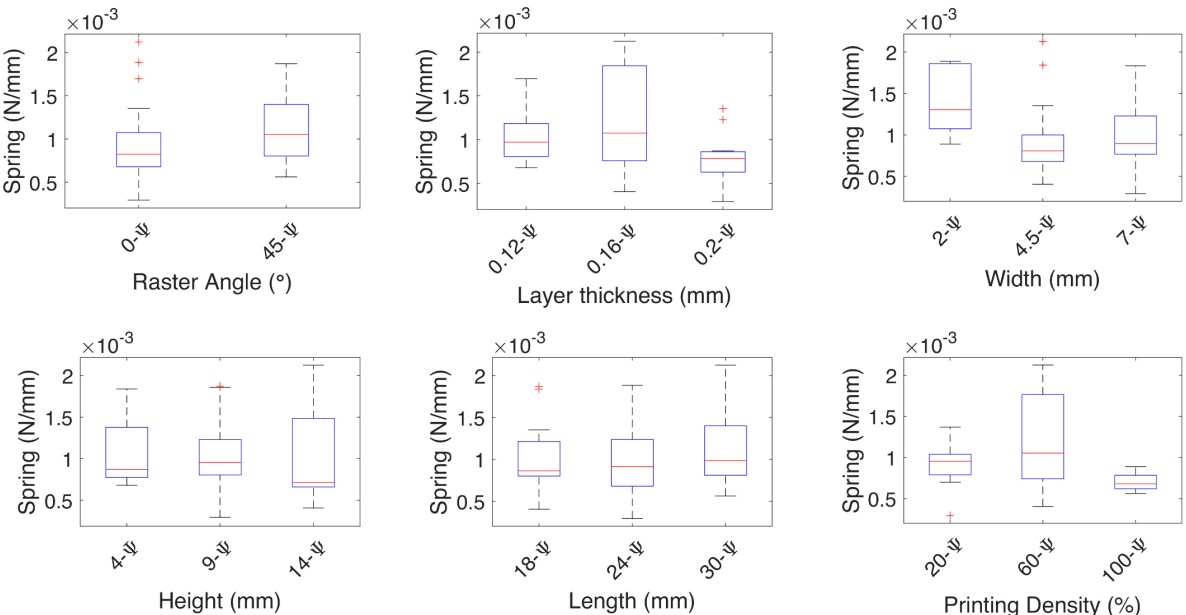

**Fig 12**. **Effect of the independent variables on the spring coefficient for angular deformations.** The effect of each independent variable on the spring coefficients are shown for the angular deformation Ψ, the medians are marked by red lines, the spring coefficient is affected by the raster angle being higher at 45° and at 0° has less variability, the geometric variables with, height, and length have the most influence on the spring coefficient.

For the spring and damping coefficients of the $z$ axis deformation, higher stiffness is observed with 45° the raster angle, and a low sensitivity to the other variables, given the medians are similar. The damping coefficient behaves in contrast to the spring coefficient, where a lower coefficient is observed with 45° the raster angle.

The damping coefficient for the $\theta$ deformation, shown in Fig 9, has higher medians as the width and height values increase. The spring coefficient shown in Fig 10 has lower medians as the width and height increase, but it increases when the length also increases.

In the case of the damping coefficient for the $\phi$ deformation, it increases with the height. The spring coefficient is affected by the raster angle, having a higher rigidity at 45°, and increasing when the height and density also increase.

In the case of the damping coefficient for the Ψ deformation, a raster angle at 45° has a higher damping coefficient; the increase in the height also increases the median and variability. The spring coefficient at 0° raster angle is higher and has less variability; the higher median is presented in the layer thickness at 0.16 $mm$, at a width of 4.5 $mm$, and a height of 9 $mm$.

To simplify the visualization of the data, a heat-map is used as shown in Fig 13, where values close to 1 show strong positive correlation, values close to -1 show strong negative correlation, and values close to 0 show little or no linear correlation. The width has significant negative correlation with the spring coefficients meaning higher width tends to decrease the stiffness of the system springs and also has a positive correlation with the damping coefficient, height is strongly correlated with the damping coefficient about the $Y$ axis reaching a value of 0.69, layer thickness has a negative correlation with the spring coefficient meaning higher layer thickness tends to reduce stiffness but increases damping. Raster angle, length, and density show weaker correlations with all outputs.

The median effect of each independent variable on the damping and spring coefficients of the TPU test specimens is shown by the red lines. The finer the layer thickness, the higher the coefficients. The raster angle mainly affects the spring constant, and it is higher at a 0° raster angle. The geometric independent variables have the same effect, where

| | RasterAngle | Layer | Width | Height | Length | Density |
|---|---|---|---|---|---|---|
| Spring x | -0.07 | -0.39 | -0.48 | -0.41 | -0.06 | -0.05 |
| Damping x | -0.03 | -0.10 | 0.32 | -0.06 | -0.08 | -0.28 |
| Spring y | 0.00 | -0.23 | -0.61 | 0.02 | -0.01 | 0.10 |
| Damping y | 0.20 | 0.32 | 0.40 | 0.69 | -0.05 | 0.02 |
| Spring z | -0.11 | -0.42 | -0.57 | -0.46 | -0.02 | -0.06 |
| Damping z | -0.15 | -0.32 | -0.15 | -0.35 | -0.02 | -0.30 |
| Spring $\theta$ | 0.07 | -0.31 | -0.58 | -0.33 | 0.07 | -0.10 |
| Damping $\theta$ | 0.23 | -0.01 | 0.05 | 0.01 | 0.09 | -0.17 |
| Spring $\phi$ | -0.04 | -0.23 | -0.44 | -0.33 | 0.19 | 0.03 |
| Damping $\phi$ | 0.02 | 0.13 | 0.16 | -0.06 | 0.24 | 0.00 |
| Spring $\Psi$ | 0.22 | -0.17 | -0.33 | -0.04 | 0.08 | 0.04 |
| Damping $\Psi$ | 0.25 | 0.10 | 0.20 | 0.20 | 0.07 | -0.01 |

**Fig 13**. **Correlation heat-map between input and output variables.** The correlation between variables are represented by values from −1 to 1 where close to 1 shows strong positive correlation, values close to −1 show strong negative correlation, and values close to 0 show little or no linear correlation.

the higher the impact on the damping coefficient, the smaller the impact on the spring coefficient, except for the height, where the smaller value contributes more.

## 4.2 Model validation

To validate the coefficients resulting from the data-based model, a Hookes-Law-based test is proposed. One end of the test specimen is fixed to a rigid support, and at the other end, a fixed load is applied as shown in Fig 14.

To measure the position and angle, an Intel RealSense D455 binocular camera and a marker at the load end are used. A 280 $g$ load is applied to the test specimens. The load is positioned at the length of the specimen and maintained for 600 $s$. The deformation of the free end of the test specimen is measured, and the angle $\theta$ and the spring constant $k_\theta$ are calculated. The moment $\tau_\theta$ is also calculated based on the applied force and length of the test specimen. The resulting spring constant $k_\theta$ is then compared to the least squares numeric result.

The error between the expected spring coefficient and the resulting spring coefficient is obtained for each of the test specimens, the mean error of the 44 spring coefficients is 0.008 $N/mm$ or 1.164% for the linear deformation $x$,

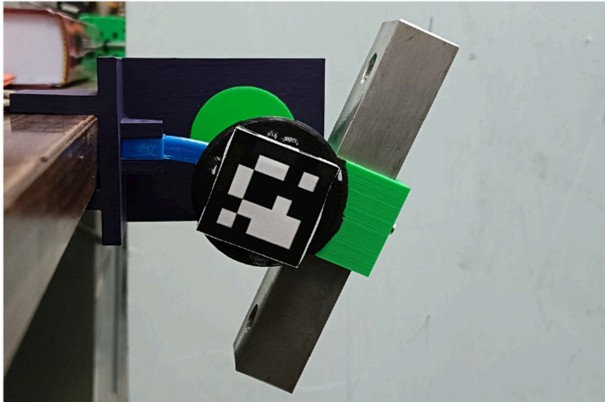

**Fig 14**. **Validation test rig.** The test specimen is fixed and a load is applied at the other end where a marker is placed.

0.0392 *N/mm* or 7.09% for the linear deformation *z*, 0.0012 *N/mm* or 4.083% for the angular deformation $\theta$, 5.5205 × $10^{-4}$ *N/mm* or 4.6004% for the angular deformation $\Psi$.

## 5 Conclusion

This paper uses the Box-Behnken design method to investigate the effect of printing and geometric parameters on a damping spring approximation of a flexible joint for robotic applications. The results show that different coefficients are obtained depending on the measure axis or angle. Increasing the printing density has the most influence on the damping instead of the spring coefficients. The raster angle has the most influence on the spring coefficients, being the highest at 0°. A finer layer thickness increases both the damping and spring effect. The width has an inverse correlation effect between its effect on damping and spring coefficients, where the correlation is positive on damping and negative on spring coefficients. The Density has the highest influence on the damping but almost no influence on the spring.

The MANOVA showed the accuracy of the model and the importance of each parameter. The P-values were below 0.05 for the layer, width, and height. Among the examined parameters, the geometry parameters have the most significant roles in modifying the damping and spring coefficients, with the height being the most significant and the length the least important.

Tuning the structural parameters of the 3D-printing process plays a significant role in the behaviour of the 3D-printed parts. The resulting low-damping factors show that the 3D-printing technology can be used in the design of robots with flexible joints to avoid the use of ball bearings. Some examples include: Soft robots that mimic the free movement of tails and tentacles, exoskeletons for motion assistance, humanoids, and quadrupeds. Future work is proposed related to the trade-offs between strength and flexibility, how joint orientation might impact fatigue life, and the influence of lattice-based sandwich structures.

## Author contributions

**Conceptualization:** Daniel Rodriguez-Flores, Carlos A. Cruz-Villar.

**Data curation:** Daniel Rodriguez-Flores.

**Formal analysis:** Daniel Rodriguez-Flores.

**Funding acquisition:** J. Enrique Chong-Quero, Carlos A. Cruz-Villar.

**Investigation:** Daniel Rodriguez-Flores, Carlos A. Cruz-Villar.

**Methodology:** Daniel Rodriguez-Flores.

**Project administration:** Héctor Cervantes-Culebro, J. Enrique Chong-Quero.

**Software:** Daniel Rodriguez-Flores.

**Supervision:** Héctor Cervantes-Culebro, Carlos A. Cruz-Villar.

**Validation:** Daniel Rodriguez-Flores.

**Visualization:** Daniel Rodriguez-Flores.

**Writing – original draft:** Daniel Rodriguez-Flores.

**Writing – review & editing:** Héctor Cervantes-Culebro, J. Enrique Chong-Quero, Carlos A. Cruz-Villar.

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
