## [Decision Letter · Decision Letter 0]

29 Jul 2025

PONE-D-25-26891Flexible joints performance assessment of additive manufacturing FDM 3D printed TPUPLOS ONE

Dear Dr. Rodriguez-Flores,

Thank you for submitting your manuscript to PLOS ONE. After careful consideration, we feel that it has merit but does not fully meet PLOS ONE’s publication criteria as it currently stands. Therefore, we invite you to submit a revised version of the manuscript that addresses the points raised during the review process.

**ACADEMIC EDITOR: As per the reviewer's feedback, I am recommending "Major Revision" of the submitted paper. If authors are willing to undergo the revision and submit a updated version along with a point-to-point response of reviewers comments, I will be happy to reconsider my decision.**

We look forward to receiving your revised manuscript.

Kind regards,

Himadri Majumder, Ph.D

Academic Editor

PLOS ONE

Journal Requirements:

[A part of this work funding was provided by CONAHCYT. The author, Héctor Cervantes-Culebro, would like to acknowledge the financial support of NOVUS (Grant number N22-302) and Writing Lab, Institute for the Future of Education, Tecnologico de Monterrey, Mexico, in the production of this work.].

[A part of this work funding was provided by CONAHCYT. The author, H´ector Cervantes-Culebro, would like to acknowledge the financial support of NOVUS (Grant number N22-302) and Writing Lab, Institute for the Future of Education, Tecnologico de Monterrey, Mexico, in the production of this work.]

[A part of this work funding was provided by CONAHCYT. The author, Héctor Cervantes-Culebro, would like to acknowledge the financial support of NOVUS (Grant number N22-302) and Writing Lab, Institute for the Future of Education, Tecnologico de Monterrey, Mexico, in the production of this work.]

5. Please remove your figures from within your manuscript file, leaving only the individual TIFF/EPS image files, uploaded separately. These will be automatically included in the reviewers’ PDF.

Reviewers' comments:

Reviewer's Responses to Questions

**Comments to the Author**

1. Is the manuscript technically sound, and do the data support the conclusions?

Reviewer #1: Partly

Reviewer #2: Partly

Reviewer #3: Yes

2. Has the statistical analysis been performed appropriately and rigorously? 

Reviewer #1: Yes

Reviewer #2: No

Reviewer #3: Yes

3. Have the authors made all data underlying the findings in their manuscript fully available?

Reviewer #1: Yes

Reviewer #2: Yes

Reviewer #3: Yes

4. Is the manuscript presented in an intelligible fashion and written in standard English?

Reviewer #1: Yes

Reviewer #2: Yes

Reviewer #3: Yes

5. Review Comments to the Author

Reviewer #1: Abstract Clarity:

The abstract provides a general overview but lacks clarity in explaining the novelty of the work. Consider explicitly stating what makes this study unique compared to existing literature on flexible joints or lattice-based sandwich structures.

I have attached comments.

Reviewer #2: 1. The title and keywords are appropriate.

2. Please improve the abstract, following the guide below:

(i) contextualization (positioning of the scenario to be evaluated), (ii) formulation of the research question, (iii) methodology, (iv) presentation of results, and (v) conclusion (scientific contribution).

3. Intro. 3rd para. Please refer to the pros and cons of the material extrusion process performance (bonding, porosity, surface roughness, hardness, compression, etc.) and better define the objectives of this research. See the following very related studies to understand what I mean.

https://doi.org/10.1007/s00170-024-14232-0 (roughness)

https://doi.org/10.1080/10426914.2024.2304843 (porosity challenges)

https://doi.org/10.1007/s00170-025-15124-7 (bonding challenges)

https://doi.org/10.1080/10426914.2023.2290258 (strength)

4. Intro. Then, please communicate the research gap more effectively and what new insights this research presents.

5. Section 1.1. Please provide a clearer explanation of the selection of the tested parameter and its levels. Why does raster angle have two levels?

6. Table 3. Please give the details of the experimental design. Factors, center point, etc. Give a reference that uses the same BBD experimental array, or explain where to find or how construct this experimental array.

7. Section 2.2 should be better communicated.

8. Explain Table 4. What does the contribution in the last column mean?

9. Conclusions should not only repeat the discussion section but also communicate this research's limits and prospects, as well as possible applications in the industry.

10. Please proofread the document carefully for units, typos, syntax, figures, details, legends, and text format.

11. The work presents some new data and, in my view, can be published after major revision.

Reviewer #3: Flexible joints performance assessment of additive manufacturing FDM 3D printed TPU

Abstract: In this study, the performance of flexible joints fabricated using Fused Deposition Modeling (FDM) 3D printing technology and TPU material is investigated for robotic applications. The objective of this study is to develop a dynamic model of these joints as a spring-damper system in order to analyze their behavior under bending and torsional loads. Using the Box-Behnken design of experiments, six parameters including print density, layer thickness, raster angle, and geometric dimensions (length, width, and height) were studied at three levels. Through mathematical modeling using the Least Squares method, the spring and damping coefficients were extracted for the fabricated samples. Subsequently, MANOVA and ANOVA analyses were employed to evaluate the significance of the effects of these parameters.

Despite the attractiveness of the topic and the comprehensiveness of the content, several suggestions are provided to enhance the overall quality of the paper. These suggestions are categorized into two main areas: the first set focuses on improving writing quality, sentence structure, and textual coherence, while the second set addresses scientific content and technical accuracy to enhance the paper’s academic level.

1. In the Abstract:

o Some sentences—particularly the second and third—are excessively long and include multiple concepts within a single complex structure, which makes comprehension and focus difficult.

o Including exact statistical error values at the end of the abstract is not particularly helpful for the general reader. It is preferable to present only a concise summary of the model’s accuracy.

o The final sentence begins with a lowercase letter ("the mean error..."), which is a grammatical error in standard English writing and should be corrected.

2. In the Introduction:

o While many references are cited, there is minimal critical analysis or synthesis of previous work. The introduction should also provide an evaluative and analytical perspective. A dedicated paragraph discussing the literature review and analyzing similar studies is recommended.

o The content lacks structural organization. There are abrupt transitions between general descriptions and technical details, which affect the overall coherence.

o Before stating the objective of the paper, it is important to clearly identify the gaps or limitations in previous research and explain how this study aims to address them.

o It is recommended to use the following papers. 4D Printing of Composite Thermoplastic Elastomers for Super‐Stretchable Soft Artificial Muscles. 3D printed elastomers with superior stretchability and mechanical integrity by parametric optimization of extrusion process using Taguchi Method. 3D printing super stretchable propylene-based elastomer.

3. In the Materials and Methods section:

o Although it is stated that the parameters were adopted from similar studies, the technical details or rationale behind the selection of specific values (e.g., why 0.12 or 0.16 mm for layer thickness?) are not explained. It is recommended to discuss the reasoning behind the choice of these parameters.

o The images showing the printing process and final printed samples are appropriate, and the schematic diagram is helpful. However, to complete this section, it is also recommended to include images of the raw materials used.

4. In the Design of Experiment section:

o The reason for selecting certain parameters, such as frequency range, is not clearly justified. The rationale behind these choices should be explicitly stated.

o The mathematical explanations (spring-damper model) are not well-integrated with the practical aspects (arm motion, hub, experiments). It would be more appropriate to present the modeling section separately.

o The application of the equations is not clearly described. It must be explicitly stated how the damping and stiffness coefficients are obtained from the experiment.

5. In the Results and discussion:

o The section referring to outlier removal should clearly explain the method used to identify these data points. For instance, it should specify whether outliers were detected based on certain statistical criteria, such as variance thresholds, interquartile range (IQR), or Z-score.

o Although the nonlinear behavior of the joints is mentioned, the proposed model is based on linear assumptions. While this may be valid for small deformations, the scientific justification for adopting a linear model should be clearly provided. For example, presenting a tensile test diagram of the samples and defining the linear operational range would help support the model's applicability.

6. In the Conclusion:

o Long and complex sentences should be rewritten into simpler ones with clearer structure to improve readability.

o The logical flow of information should be maintained (objective, method, results, interpretation) to enhance coherence.

o Statistical interpretations (e.g., p-values) should be presented more precisely and understandably, so that readers can clearly grasp the significance of the findings.

6. PLOS authors have the option to publish the peer review history of their article (what does this mean?). If published, this will include your full peer review and any attached files.

Reviewer #1: **Yes: **Ali Sadeghianmaryan

Reviewer #2: No

Reviewer #3: No

---

## [Author Response · Author response to Decision Letter 1]

15 Sep 2025

We trust that the revised manuscript now meets the reviewers’ expectations. We again thank the reviewers for their valuable insights, which significantly improved the quality of this work.

---

## [Decision Letter · Decision Letter 1]

3 Oct 2025

PONE-D-25-26891R1Flexible joints performance assessment of additive manufacturing FDM 3D printed TPUPLOS ONE

Dear Dr. Rodriguez-Flores,

Thank you for submitting your manuscript to PLOS ONE. After careful consideration, we feel that it has merit but does not fully meet PLOS ONE’s publication criteria as it currently stands. Therefore, we invite you to submit a revised version of the manuscript that addresses the points raised during the review process.

We look forward to receiving your revised manuscript.

Kind regards,

Himadri Majumder, Ph.D

Academic Editor

PLOS ONE

Journal Requirements:

Reviewers' comments:

Reviewer's Responses to Questions

**Comments to the Author**

1. If the authors have adequately addressed your comments raised in a previous round of review and you feel that this manuscript is now acceptable for publication, you may indicate that here to bypass the “Comments to the Author” section, enter your conflict of interest statement in the “Confidential to Editor” section, and submit your "Accept" recommendation.

Reviewer #2: (No Response)

Reviewer #3: (No Response)

2. Is the manuscript technically sound, and do the data support the conclusions?

Reviewer #2: Partly

Reviewer #3: (No Response)

3. Has the statistical analysis been performed appropriately and rigorously? 

Reviewer #2: I Don't Know

Reviewer #3: (No Response)

4. Have the authors made all data underlying the findings in their manuscript fully available?

Reviewer #2: No

Reviewer #3: (No Response)

5. Is the manuscript presented in an intelligible fashion and written in standard English?

Reviewer #2: Yes

Reviewer #3: (No Response)

6. Review Comments to the Author

Reviewer #2: A. In the revised manuscript, the following concern is not adequately explained by the authors. Please, Table 3 is Table 3, not Table 2. Explain how to implement BBD given a parameter with 2 levels. Also, explain better the infill type and why 0 and 90 degrees are the same.

1.16 Reviewer # 2, Concern # 6

Table 3. Please give the details of the experimental design. Factors, center point, etc.

Give a reference that uses the same BBD experimental array, or explain where to find or

how construct this experimental array.

B. Please explain the following sentense.

'The results of the 160 test specimens were filtered and cleaned, eliminating zeroes, repeated data, and outliers, resulting in only 44 usable datasets.'

Reviewer #3: Dear Authors,

The manuscript (Flexible joints performance assessment of additive manufacturing FDM 3D printed TPU) is well-revided and can be accepted in present form.

7. PLOS authors have the option to publish the peer review history of their article (what does this mean?). If published, this will include your full peer review and any attached files.

Reviewer #2: No

Reviewer #3: **Yes: **Davood Rahmatabadi

---

## [Author Response · Author response to Decision Letter 2]

15 Oct 2025

Thank you for allowing a resubmission of our manuscript, with an opportunity to address the reviewers’

comments.

We are uploading (a) A rebuttal letter that responds to each point raised by the academic editor and

reviewers, labeled ’Response to Reviewers’. (b) A marked-up copy of the manuscript that highlights

changes made to the original version, labelled ’Revised Manuscript with Track Changes’. (c) An unmarked

version of the revised paper without tracked changes labelled ’Manuscript’.

We trust that the revised manuscript now meets the reviewers’ expectations. We again thank the

reviewers for their valuable insights, which significantly improved the quality of this work.

Best Regards,

Daniel Rodríguez Flores, et al.

---

## [Decision Letter · Decision Letter 2]

26 Oct 2025

Flexible joints performance assessment of additive manufacturing FDM 3D printed TPU

PONE-D-25-26891R2

Dear Dr. Rodriguez-Flores,

We’re pleased to inform you that your manuscript has been judged scientifically suitable for publication and will be formally accepted for publication once it meets all outstanding technical requirements.

Kind regards,

Himadri Majumder, Ph.D

Academic Editor

PLOS ONE

Additional Editor Comments (optional):

Reviewers' comments:

Reviewer's Responses to Questions

**Comments to the Author**

1. If the authors have adequately addressed your comments raised in a previous round of review and you feel that this manuscript is now acceptable for publication, you may indicate that here to bypass the “Comments to the Author” section, enter your conflict of interest statement in the “Confidential to Editor” section, and submit your "Accept" recommendation.

Reviewer #2: All comments have been addressed

2. Is the manuscript technically sound, and do the data support the conclusions?

Reviewer #2: Yes

3. Has the statistical analysis been performed appropriately and rigorously? 

Reviewer #2: Yes

4. Have the authors made all data underlying the findings in their manuscript fully available?

Reviewer #2: Yes

5. Is the manuscript presented in an intelligible fashion and written in standard English?

Reviewer #2: Yes

6. Review Comments to the Author

Reviewer #2: Authors have adequately addressed all my recommendations in the revised manuscript. Therefore I recommend this work for publication.

7. PLOS authors have the option to publish the peer review history of their article (what does this mean?). If published, this will include your full peer review and any attached files.

Reviewer #2: No

---

## [Editor Report · Acceptance letter]

PONE-D-25-26891R2

PLOS ONE

Dear Dr. Rodriguez-Flores,

I'm pleased to inform you that your manuscript has been deemed suitable for publication in PLOS ONE. Congratulations! Your manuscript is now being handed over to our production team.

Kind regards,

on behalf of

Dr. Himadri Majumder

Academic Editor

PLOS ONE